# Effects of government policies and the Nowruz holidays on confirmed COVID-19 cases in Iran: An intervention time series analysis

Ali Hadianfar[1,2], Razieh Yousefi[1,2], Milad Delavary[3], Vahid Fakoor[4]*, Mohammad Taghi Shakeri[5], Martin Lavallière[3]

**1** Student Research Committee, Mashhad University of Medical Sciences, Mashhad, Iran, **2** Department of Biostatistics, School of Health, Mashhad University of Medical Sciences, Mashhad, Iran, **3** Department of Health Sciences, Laboratoire BioNR and Centre Intersectoriel en Santé Durable (CISD), Université du Québec à Chicoutimi, Chicoutimi, Québec, Canada, **4** Department of Statistics, Faculty of Mathematical Sciences, Ferdowsi University of Mashhad, Mashhad, Iran, **5** Social Determinants of Health Research Center, Mashhad University of Medical Sciences, Mashhad, Iran

* Fakoor@um.ac.ir

**Data Availability Statement:** All relevant data are within the paper and its Supporting Information files.

## Abstract

### Background

Public health policies with varying degrees of restriction have been imposed around the world to prevent the spread of coronavirus disease 2019 (COVID-19). In this study, we aimed to evaluate the effects of the implementation of government policies and the Nowruz holidays on the containment of the COVID-19 pandemic in Iran, using an intervention time series analysis.

### Methods

Daily data on COVID-19 cases registered between February 19 and May 2, 2020 were collected from the World Health Organization (WHO)'s website. Using an intervention time series modeling, the effect of two government policies on the number of confirmed cases were evaluated, namely the closing of schools and universities, and the implementation of social distancing measures. Furthermore, the effect of the Nowruz holidays as a non-intervention factor for the spread of COVID-19 was also analyzed.

### Results

The results showed that, after the implementation of the first intervention, i.e., the closing of universities and schools, no statistically significant change was found in the number of new confirmed cases. The Nowruz holidays was followed by a significant increase in new cases (1,872.20; 95% CI, 1,257.60 to 2,476.79; p<0.001)), while the implementation of social distancing measures was followed by a significant decrease in such cases (2,182.80; 95% CI, 1,556.56 to 2,809.04; p<0.001).

**Funding:** The author(s) received no specific funding for this work.

**Competing interests:** The authors have declared that no competing interests exist.

## Conclusion

The Nowruz holidays and the implementation of social distancing measures in Iran were related to a significant increase and decrease in COVID-19 cases, respectively. These results highlight the necessity of measuring the effect of health and social interventions for their future implementations.

## Introduction

As a global pandemic, COVID-19 has resulted in 403,080 deaths [1, 2] and 7,028,020 confirmed cases as of June 7, 2020. The first confirmed COVID-19 cases in Iran were reported in Qom on February 19, 2020. Shortly after this, COVID-19 cases were reported in other Iranian cities, and the country is still heavily impacted by this pandemic as of June 8, 2020 with 175,927 confirmed cases and 8,351 deaths [1].

COVID-19 can lead to severe acute respiratory distress syndrome (ARDS), anemia, secondary infection, acute cardiac injury, fever, fatigue, dry cough, and ultimately, to death [3, 4]. Due to these serious symptoms, countries have implemented COVID-19-related policies to limit the spread of the disease and prevent the exhaustion of the national health system's resources and capacities [5]. However, there are a few studies about the impact of such government policies on the number of COVID-19 cases, and questions remain about the impact of such measures on case numbers. Siedner et al. used intervention time series analysis to investigate whether the implementation of social distancing measures was associated with a reduction in the mean daily growth rate of COVID-19 cases in US states. Their results showed that social distancing measures were associated with a decrease in pandemic growth [6].

Time series analysis has been used to model trends in the prevalence and incidence of COVID-19 cases registered in the Johns Hopkins epidemiological database (https://coronavirus.jhu.edu/) [7]. Similarly, Soudeep et al. proposed a time series model to analyze the trend pattern of COVID-19 incidence [8], and Petropoulos et al. provided statistical forecasts for confirmed cases of COVID-19, using robust time series models [9]. In Iran, Moftakhar and Seif predicted the number of newly infected patients using the ARIMA model on March 20, 2020, anticipating 3,574 cases by April 20, 2020 [10]. Jamshidi et al. applied a model for COVID-19 prediction in Iran based on China's parameters. According to their prediction, the expected cumulative number of confirmed cases in Iran could have reached 29 000 from March 25 to April 15, 2020 [11]. Time series analyses can also detect change points and assess the influence of interventions. Change points are abrupt changes that represent transitions occurring in time series data [12]. An interventional analysis is useful when the exact effect of interventions is of interest. In other words, the analysis aims to predict or identify an intervention and its related effects using data, by applying a time series analysis [13].

In Iran, the government implemented health policies for COVID-19 and applied social distancing rules to limit its transmission. Amid the pandemic, Iran celebrated the Nowruz holidays (the New Year in Iran, March 20 to May 2, 2020), a time when people usually visit elderly relatives. This elderly population represents the most at-risk population for severe disease and death if infected with COVID-19 [14]. It was expected that the outbreak of COVID-19 would encourage people to stay at home. Despite all warnings, Iranians started their Nowruz travels inside the country, leading to a high incidence of COVID-19 disease in the Northern provinces [15]. In their study, Heidari and Sayfouri showed that Persian Nowruz aggravated the COVID-19 crisis in Iran [16]. Besides social distancing measures, the effect of national holidays in countries affected by COVID-19 has not been well-researched.

Evidence suggests that multiple factors can influence the number of COVID-19 cases in Iran. We employed a time series analysis to analyze the trend of new COVID-19 cases in Iran following the establishment of social distancing directives and related government policies and the Nowruz holidays (Iranian New Year, also known as the Persian New Year that usually occurs on March 21[st]) to case numbers.

## Materials and methods

The collected dataset included the new confirmed cases of COVID-19 in Iran from February 19 to May 2, 2020. The dataset was obtained from the daily reports of the Iranian Ministry of Health and Medical Education (https://behdasht.gov.ir/), which is identical to the COVID-19 data published on the WHO website (https://www.who.int/emergencies/diseases/novel-coronavirus-2019/situation-reports/), which aggregates case data from national authorities. Summary of statistics including mean, standard deviation, minimum, maximum, skewness and kurtosis are presented in Table 1.

### Interventions

In Iran, different health policies have been implemented to control the spread of COVID-19 [5, 17]. The first intervention enacted in Iran was the closing of kindergartens, schools, and universities. Although emphasis was put on the importance of handwashing, wearing a mask, and staying at home, most of the population did not take these measures seriously. The second intervention that was comprised of new social distancing measures was launched on March 27, 2020, and the police were in charge of enforcing this. The entry of traffic into the cities was restricted to their residents, only. People were required to return to their homes, as many Iranians had traveled to other provinces during the Nowruz holidays. Also, all new travel outside of the cities was banned for non-essential purposes, and an automatic vehicle seizure of 23 days, and 5,000,000 IRR fines (approximately US$ 35) were imposed for travel ban offenders [17]. On April 17, 2020 Iran decreased social distancing measures and focused on the Smart Social Distancing Plan [18]. This plan was in line with social distancing and provided conditions for society to gradually return to normal [16]. After May 2[nd], Iran reached the disease management phase, expanded its active case finding program, implemented contact tracing, and tested those in close contact with COVID-19 patients.

### Statistical analysis

An autoregressive integrated moving average (ARIMA) is a powerful tool to forecast a time series model [19], and a seasonal auto-regressive integrated moving average (SARIMA) model is used when a seasonal component is involved [20]. SARIMA, firstly, proposed by Box and Jenkins in the 1970s [19]. It is presented as SARIMA $(p, d, q)(P, D, Q)_S$, where p is the order of auto-regressive (AR), q is the order of moving average (MA), d is the order of the differences.

**Table 1. Descriptive statistics of new confirmed cases per day of COVID-19.**

| Statistics | Estimate |
|---|---|
| Mean | 1303.4 |
| Std. dev | 845.2 |
| Min | 2.0 |
| Max | 3186.0 |
| Skewness | 0.472 |
| Kurtosis | 2.67 |

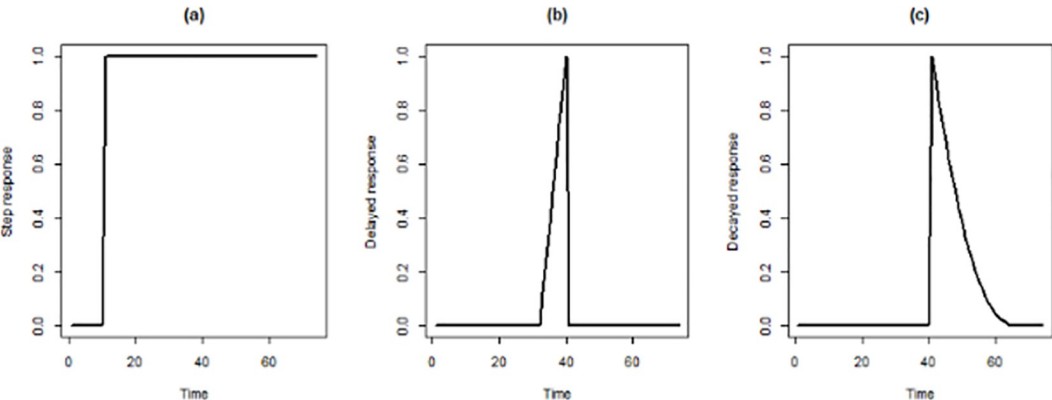

**Fig 1.** Three type of interventions used in dynamic regression; step response (a), delayed response (b), and decayed response (c).

The ACF and PACF are used for knowing the order of AR and MA. Also, based on the trend and season of time series, the order of differences will be recognized. Moreover, P, D, and Q are the corresponding seasonal orders, with $_s$ as the steps of the seasonal differences [19]. The effect of the intervention, a one-off event affecting the new confirmed cases variable, was analyzed by using intervention time series analysis. Introduced by Box and Tiao, intervention time series analysis is an approach for handling the effectiveness of interventions in a dynamic regression framework [21]. Although it is assumed that an intervention can only happen at a specific time, its effects can spread over time. Fig 1 indicates examples of these data points, over time. In this study, three types of interventions, including step, delayed (linear trend), and decayed (exponential trend) response, were considered to evaluate the impact of the Nowruz holidays and government policies (Fig 2).

Fig 3 shows the time series of newly confirmed COVID-19 cases in Iran from February 19 to May 2, 2020. Two government policies were considered as two separate interventions. The first policy was the closing of schools and universities (CSU), which was implemented on March 1, 2020. And on March 27, 2020, when new legislation came into force requiring the social distancing measures (SDM) to be implemented (second policy) and enforced by the police.

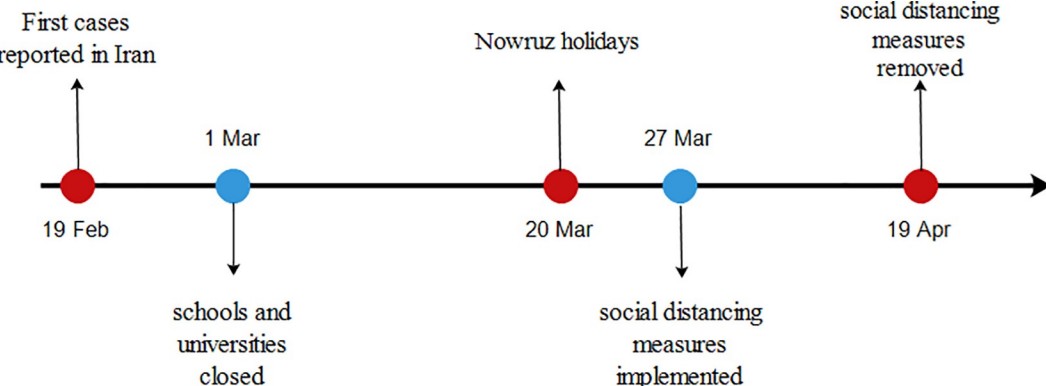

**Fig 2. Timeline of COVID-19-related events in Iran, from February 19 to April 19, 2020.** Red dots present events in the COVID-19 outbreak, blue dots present control measures implemented by the Iranian government.

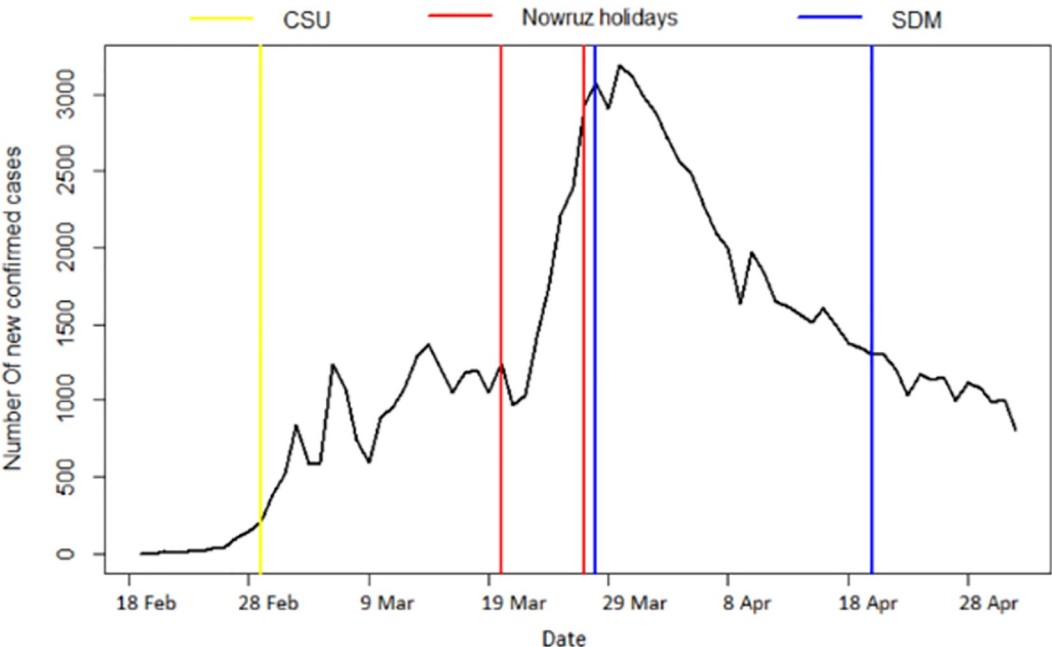

**Fig 3. The times series of new confirmed COVID-19 cases in Iran from February 19, 2020 to May 2, 2020.** Yellow, Red and Blue vertical lines indicate time of implementation of Closing schools and universities (CSU), Nowruz holidays and implementation of new social distancing measures (SDM), respectively.

For time series modeling, it is assumed that the intervention occurs at a time point, say '$\tau$', where a dummy variable can be considered 0 before the intervention and 1 after the intervention [22]. This is called a step intervention. Fig 3 shows the effect of the CSU and SDM on the series in x using these dummy variables with $\tau_1$ and $\tau_2$ corresponding to March 1, 2020 and March 27, 2020, respectively. The model is then,

$$Y_t = \beta_0 + \beta_1 CSU_t + \beta_2 Nowruz\ holidays_t + \beta_3 SDM_t + e_t \tag{1}$$

, where $Y_t$ represents the outcome variable overtime point $t$, which is considered the number of confirmed COVID-19 cases. The value of $\beta_0$ is the baseline level of the response variable (also the initial value at $t = 0$). $\beta_1$ and $\beta_3$ represent the effects of the drop in newly confirmed COVID-19 cases because of the CSU and SDM interventions. Also, $\beta_2$ represents the effect of the Nowruz holidays in increasing the number of new confirmed COVID-19 cases. Furthermore, a SARIMA model was used for the error term $e_t$ that must follow the white noise. We computed 95% confidence intervals based on Z test.

Plots of the autocorrelation function (ACF), partial autocorrelation function (PACF), and Ljung–Box test were proposed for determining uncorrelated residuals [23]. Furthermore, residual plots were used to assess the zero-mean assumption, and the normality of residuals was evaluated using the Kolmogorov-Smirnov test. All the model developments, computations, and comparisons were performed using the R forecast package, and the statistical significance level was set at P-value less than 0.05.

## Ethical statement

The data was provided by the Iranian Ministry of Health and is publicly available online on the WHO website. Therefore, ethical approval was not required.

**Table 2. The effects of the government policies and Nowruz holidays on the daily confirmed new COVID-19 cases, SARIMA models.**

| Output | Estimate | SE | P–value | 95%CI[c] | Ljung–Box | Kolmogorov–Smirnov |
|---|---|---|---|---|---|---|
| Intercept | 645.47 | 289.36 | 0.03 | (78.32, 1212.62) | | |
| Effect (CSU[a]) | 130.0 | 172.87 | 0.45 | (-208.82, 468.83) | | |
| Nowruz holidays | 1872.20 | 313.57 | <0.001 | (1257.603, 2486.79) | | |
| Effect (SDM[b]) | 2182.80 | 319.51 | <0.001 | (1556.56, 2809.04) | | |
| Noise | (1,0,0)(1,0,0) | | | | 0.18 | 0.96 |

[a] Closing schools and universities (CSU), March 1, 2020

[b] New social distancing measures (SDM), March 27, 2020

[c] Confidence interval (CI) were obtained using a Z test.

## Results

After detecting the first COVID-19 case on February 19, 2020 in Iran, the daily number of new confirmed cases rose gradually to 1,046 until March 19, 2020. With the beginning of the Nowruz holidays on March 20, the number of new confirmed cases increased sharply with a three-day delay, exceeding 3,000 at the maximum point on March 30 (see Fig 3 above).

As shown in Table 2, the mean number of new confirmed cases in Iran was 645.47 cases per day (95% CI, 78.32 to 1,212.62; p = 0.03) before March 2, 2020 (before the interventions). A significant linear increment in new confirmed cases was observed, which was about 0 to 1,872.20 (95% CI, 1,257.603 to 2,486.79; p<0.001), after the Nowruz holidays. Furthermore, with a three-day delay, a linear increase in COVID-19 case numbers was observed for eight days after the beginning of the Nowruz holidays.

The results showed no significant change in the number of new confirmed cases after the implementation of the first intervention (CSU). Whereas, after implementing the second intervention (SDM), new daily confirmed cases decreased significantly (p<0.001) from 2,182.80 (95% CI, 1,556.56 to 2,809.04) to 1,343 during the intervention period.

The new confirmed COVID-19 cases model is as follows:

$$Y_t = 645.47 + 130\ CSU_t + 187.2\ Nowruz\ holidays_t + 218.8\ SDM_t + e_t \qquad (2)$$

, where $e_t \sim SARINA(1,0,0)(1,0,0)_{14}$ and indicates a seasonal variation in the new daily cases of COVID-19 in Iran. There is a recurrent pattern of changes in the number of new cases within the period series. This season was 14 days, which shows that a uniform pattern happens every 2 weeks. This can be due to the disease's incubation period, which has been documented at a maximum of 14 days [24].

It seems that the effect of social distancing rules had been significant, considering the gradual downtrend in the daily number of new cases. As shown in Fig 3, the daily number of cases exponentially decreased after March 27 for 23 days. In detail, this reduction was observed from March 27 to April 19, 2020, when the government removed the social distancing restriction.

For model diagnostics, the residuals should be white noise. In this connection, there was no pattern in the plot of residuals, and they were randomly scattered around zero. Also, there were no spikes in the autocorrelation and partial autocorrelation functions, indicating that there was no remaining autocorrelation regarding the residuals (Fig 4). Furthermore, the Ljung-Box (LB) test was utilized to understand whether any of the groups of autocorrelations of a time series are different from zero. As shown in Table 2, uncorrelated residuals were confirmed at the 5% significance level (p>0.05). Moreover, the Kolmogorov-Smirnov test established the normality of residuals. The goodness of fit statistics means absolute percentage error

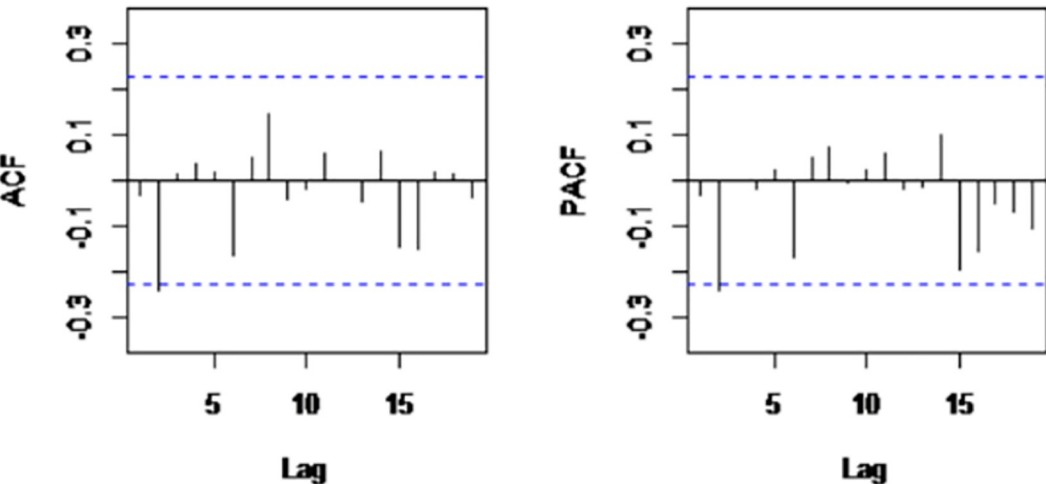

**Fig 4. Autocorrelation and partial autocorrelation functions of SARIMA residuals.**

obtained was 1.88, which is highly accurate forecasting, based on the research study by Lewis [25].

Table 3 shows the prediction of Covid-19 new confirmed cases, with 95% confidence interval, in 7 days. It should be noted that the prediction can be utilized if the observed spreading pattern, and the number and type of test for detection the COVID-19 cases continues as before and if policies and restrictions are not removed. Therefore, the number of daily new cases would be 813 and 814 for May 3rd 2020 and May 4th 2020, respectively.

## Discussion

COVID-19 is an infectious disease spread through direct contact between individuals [26]. Outbreak control measures implemented to diminish the contacts within the population can reduce the height of the peak, the speed at which the virus spreads, and the final scope of the pandemic. In Iran, different policies and strategies have been implemented, based on the experience and recommendations of China and the WHO to control the outbreak of COVID-19 [5, 17]. In general, we found a correlation between the national Nowruz holidays, the new social distancing measures and the number of newly confirmed COVID-19 cases in Iran. However, the closing of kindergartens, schools, and universities was not followed by a reduction in new cases.

**Table 3. Forecasted number of daily Covid-19 new cases with 95% confidence intervals.**

| Days (of 2020) | Prediction | | |
|---|---|---|---|
| | SARIMA | 95% CI for SARIMA | |
| | | Lower | Upper |
| 3-May | 813 | 516 | 1110 |
| 4-May | 814 | 410 | 1217 |
| 5-May | 829 | 354 | 1304 |
| 6-May | 815 | 287 | 1343 |
| 7-May | 769 | 200 | 1338 |
| 8-May | 822 | 221 | 1423 |
| 9-May | 818 | 190 | 1445 |

The first government policy implemented by Iran to combat the outbreak of COVID-19 was the closing of kindergartens, schools, and universities. The result of our study shows that this intervention did not contribute to control the pandemic. A review study, including 16 studies about the effect of school closures during coronavirus outbreaks, indicates that their impact on the spread of COVID-19 is very weak [27]. Therefore, policymakers need to be aware of the uncertainty of evidence about the efficacy of school closures to slow down COVID-19, and may need to consider combinations of various distancing measures, instead.

The Nowruz holidays (an Iranian national holidays) created a significant increase in the number of COVID-19 cases, three days after its start. Due to the lack of personal protective equipment such as masks and disinfectants, as well as a high level of contact between people, the Nowruz holidays was an occasion for easy transmission of the disease in Iran [28].

Prior to the implementation of social distancing measures, the transmission rates of the COVID-19 infection in Iran were increasing. New social distancing measures were implemented in Iranian Provinces to reduce the risk of expansion of the pandemic. The most important part of the social distancing rules was travel restrictions and car seizure as well as 5,000,000 IRR (US$ 35) fines implemented on March 27, 2020. Our findings indicated that the implementation of the social distancing measures in Iran were effective in controlling the spread of the outbreak, and that the number of new daily COVID-19 cases significantly decreased after adopting these measures. Our results regarding the impact of social distancing policies on the number of COVID-19 cases support earlier findings on the effectiveness of such measures [6, 29, 30]. These studies have shown a decrease in the average daily new COVID-19 cases, once sanitary measures were implemented.

There are some potential reasons for the ineffectiveness of government policies to reduce the number of new cases. In Iran, like some other countries such as China [31], the outbreak coincided with a national holiday. We found a negative relationship between the Nowruz holidays and the number of cases, which might have dwindled the effectiveness of disease control measures. The general population was requested to stay at home and self-quarantine during the Nowruz holidays, as well as refrain from visiting their families. Therefore, the degree of the outbreak was expected to be manageable. This somewhat contradictory result may be because millions of Iranians traveled around the country. With the beginning of the Nowruz holidays, the police reported heavy traffic toward northern cities, therefore traveling might have exacerbated the spread of the outbreak. This finding corroborates the results of Heidari and Sayfouri, who suggested that Nowruz aggravated the COVID-19 crisis in Iran [16]. It should be noted that, by increasing the number of tests, diagnoses of COVID-19 were increased. It is valuable to mention that the rate of test increment was non-stop, even after social distancing. Therefore, the decrease of COVID-19 patients after the enforcement of social distancing cannot be attributable to a lack of access to testing or to improper distribution.

It is worth noting that, due to the rapidly increasing incidence trend of COVID-19, it is not only essential to design and implement rules but also to critically plan the moment of implementation of such measures. Late implementation of social distancing measures, such as in Italy, can lead to an exponential increase in the mortality rate in the population [32]. Previous studies have also indicated that earlier implementation of measures can be more productive. A recent study has shown that every one-day delay in the implementation of social distancing measures leads to a 2.41-day delay in containment of the pandemic [15]. The impact of delays may be particularly significant for communities that are prone to rapid disease transmission. For example, during the Nowruz holidays in Iran, people visited multiple relatives and many others used the two-week break to travel to tourist destinations across the country. Therefore, earlier implementation of restriction rules and prevention of non-essential travel could have made it easier to control the spread of the outbreak.

There are some limitations to the analysis conducted in this study. First, we only analyzed the available data related to the period from February 19 to May 2, 2020, as the Iranian Ministry of Health began its active case finding program during this period. Second, we did not have access to sub-groups and geographical data. These data could be valuable in determining the heterogeneity of the effect of social distancing in different subgroups and for different geographical areas. Moreover, there were also some variables, such as the level of access to the healthcare system, the changing diagnostic criteria of COVID-19, the people's compliance to health rules, and preventive programs that could influence the effectiveness of social distancing.

Notwithstanding these limitations, the results of this study suggest that in general, government policies can have a significant influence on COVID-19 case numbers and can be used to control similar outbreaks. Our results also highlight the critical influence of national holidays on the spread of COVID-19 in Iran. Our findings should be considered for planning and implementing future measures. Future studies should also address the cost-benefit of these plans and other possible options when deciding on the implementation of such national measures.

## Conclusion

This study evaluated the effects of two government policies and the Nowruz holidays on the number of new COVID-19 cases in Iran, using intervention time series analysis. The results indicated that the Nowruz holidays significantly increased case numbers. We also found that the implementation of social distancing measures as a non-pharmaceutical, and non-medical, intervention in Iran had a significant influence on reducing the new daily cases of COVID-19 and could effectively control the spread of the disease in Iran.

## Supporting information

**S1 Data.**
(XLSX)

## Acknowledgments

The authors would like to thank Mr. Felix Siebert from the Technische Universität of Berlin for his substantial contributions in revising the manuscript.

## Author Contributions

**Conceptualization:** Vahid Fakoor, Martin Lavallière.

**Data curation:** Ali Hadianfar.

**Formal analysis:** Milad Delavary.

**Investigation:** Razieh Yousefi, Mohammad Taghi Shakeri.

**Methodology:** Vahid Fakoor.

**Software:** Milad Delavary.

**Supervision:** Vahid Fakoor.

**Writing – original draft:** Ali Hadianfar.

**Writing – review & editing:** Razieh Yousefi, Mohammad Taghi Shakeri, Martin Lavallière.

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
