## [Decision Letter · Decision Letter 0]

12 Feb 2021

PONE-D-20-30409

Effects of government policies and the Nowruz holidays on confirmed COVID-19 cases in Iran: An intervention time series analysis

PLOS ONE

Dear Dr. Vahid Fakoor

Thank you for submitting your manuscript to PLOS ONE. After careful consideration, we feel that it has merit but does not fully meet PLOS ONE’s publication criteria as it currently stands. Therefore, we invite you to submit a revised version of the manuscript that addresses the points raised during the review process.

We look forward to receiving your revised manuscript.

Kind regards,

Cesar V Munayco, M.D, MSc, MPH, DrPH

Academic Editor

PLOS ONE

Journal Requirements:

3. We note that there is a discrepancy within the manuscript regarding the dates from which data were collected, for instance between the abstract and the methods section. Please amend as necessary.

Additional Editor Comments:

This manuscript is interesting, but there are some issues in interpreting the results and their limitations. The authors must be sober and cautious when they value their findings.

It would be better if the authors would consider a counterfactual because there is no way to confirm that this epidemic trend will behave the same without these interventions. Most of the time, some areas do not fully accomplish all interventions, and these areas are perfect as a counterfactual.

The authors must give information about the preemptive measures' compliance to value them and evaluate their effect on the epidemic curve.

Reviewers' comments:

Reviewer's Responses to Questions

**Comments to the Author**

1. Is the manuscript technically sound, and do the data support the conclusions?

Reviewer #1: Partly

2. Has the statistical analysis been performed appropriately and rigorously? 

Reviewer #1: No

3. Have the authors made all data underlying the findings in their manuscript fully available?

Reviewer #1: Yes

4. Is the manuscript presented in an intelligible fashion and written in standard English?

Reviewer #1: Yes

5. Review Comments to the Author

Reviewer #1: The authors have conducting fairly straightforward analyses using highly aggregated surveillance data with known limitations to answer complex questions. Greater attention needs to be placed in the analytical approach and its appropriateness to the research question. Conceptually, authors need to better address the nature of epidemic growth and reduction, particularly taking into account the limitations of the data used. Finally, the authors make numerous statements attributing causality to their findings without carefully consideration of the causality principles and how their evidence supports them.

The ARIMA model intrinsically does not have growth restriction and in theory it could grow without limits, which does not correspond to reality, because during epidemics, even in large populations the number of susceptible individuals is eventually exhausted. This is conceptually erroneous when applied to the complete growth and reduction cycle of a pandemic, and it assumes that all the reduction cycle is due to the physical distancing measures, without taking into account the reduction of the susceptible individuals and its impact. It should be clarified if the autocorrelation is the only mechanism that accounts for the multiplicative growth pattern of the epidemic. Particularly because the use of pulse-like effects to estimate the potential effect of the control measures and the Norwuz holidays does not affect the autocorrelation or the slope of the curve.

Likewise, it should be clarified if the ARIMA model uses a Poisson or another distribution for count data such as cases, and what are the effects of that in the estimations.

Similarly, the authors do little to explore other potential explanations that could lead to the temporal correlations observed. Rarely a single measure takes place alone, and often there are multiple simultaneous efforts to mount a public health response that could also lead to the trends observed.

What is the precision of estimates, and is model fitness high enough that can serve to respond trend questions? No goodness of fit statistics are presented nor the observed and estimated results are presented, visually or in a tabulation.

Finally, no sensitivity or sub-group analyses are presented to document the robustness of the results and related conclusions. Most countries have shown highly heterogeneous sub-national epidemic expansions in the first wave of the pandemic, with the national pattern primarily reflecting trends in the capital or largest cities, and very different pattern in smaller or distant/less connected cities. Conclusions could greatly change when addressing these heterogeneity.

Conceptually, which of the three types of interventions better fits the interventions evaluated, and why were the others considered? Did the empirical results of the regression matched the expectation due to the conceptual type of intervention?

Could the closure of schools have a relatively smaller effect than expected and was not detectable with the methods used? What if this measure slowed down the decrease observed in mid-March 2020?

The abstract should present actual results to support the authors’ statements and conclusions, as p-values are insufficient for that purpose.

The case definition needs to be clearly described stating how testing availability and distribution countrywide affects the validity and proper interpretation of findings

How many days are expected to pass for interventions to start being effective? How was this lag introduced in the analyses?

6. PLOS authors have the option to publish the peer review history of their article (what does this mean?). If published, this will include your full peer review and any attached files.

Reviewer #1: No

---

## [Author Response · Author response to Decision Letter 0]

14 Apr 2021

Professor. Cesar V Munayco

Academic Editor

PLOS ONE 

SUBJECT: Resubmission of manuscript PONE-D-20-30409

Re: Effects of government policies and the Nowruz holiday on confirmed COVID-19 cases in Iran: An intervention time series analysis

Dear Editor,

We would like to thank you the attention brought to our manuscript. Also, we would like to thank the Associate Editor and the Reviewer for their careful reading of our manuscript and their insightful comments. We have reviewed our manuscript accordingly. The comments have been noted and we have done our best to revise the paper accordingly. Below is a point-by-point response to each comment raised by the Associate Editor and the Reviewer (original comments from reviewers appear in red, responses in black). It should be noted that the modifications in the manuscript are highlighted using track changes in Word. In addition, a final version of the manuscript, including all types of corrections, is attached. 

We hope that the revised version of the manuscript will now meet the requirements of PLOS ONE.

Yours sincerely,

Vahid Fakoor

Associate Professor, Department of Statistics, Faculty of Mathematical Sciences, Ferdowsi University of Mashhad, Mashhad-Iran,

P.O.Box, 9177943369 

Mobile: 0989151038414

E-mail: fakoor@um.ac.ir

Academic Editor:

We would like to thank you for your comments. Below please see our response to the comments and suggestions for which we have revised our manuscript.

Academic Editor’s Comments to the Author

Point-by-point responses to the issues raised by the Academic Editor:

1) raises questions about

and https://journals.plos.org/plosone/s/file?id=ba62/PLOSOne_formatting_sample_title_authors_affiliations.pdf. 

ANSWER: The manuscript has been carefully modified in accordance with the PLOS ONE’s style requirements.

• We suggest you thoroughly copyedit your manuscript for language usage, spelling, and grammar. If you do not know anyone who can help you do this, you may wish to consider employing a professional scientific editing service. 

A clean copy of the edited manuscript (uploaded as the new *manuscript* file).

ANSWER: Thanks for your suggestion. We have sought the help of a professional editing service to review our manuscript and the documents requested are attached.

The editing service was provided by:

Micheline Harvey

Translator 

Traduction – révision – correction – transcription/Translation – revision – correction – transcription

418-806-6794

www.secretairevirtuelle.com

Membre du Réseau des professionnels en soutien administratif virtuel RPSAV

• We note that there is a discrepancy within the manuscript regarding the dates from which data were collected, for instance between the abstract and the methods section. Please amend as necessary.

ANSWER: Thank you for your comment. We amended it in line 21, as required.

• Please amend either the abstract on the online submission form (via Edit Submission) or the abstract in the manuscript so that they are identical

ANSWER: Thank you for your comment. We amended it, as required.

Additional Editor Comments:

• This manuscript is interesting, but there are some issues in interpreting the results and their limitations. The authors must be sober and cautious when they value their findings.

ANSWER: Thank you for your constructive suggestion. We have modified some parts of the results section in lines 168-173, as follows:

As shown in Table 1, the mean number of new confirmed cases in Iran was 645.47 cases per day (95% CI, 78.32 to 1,212.62; p=0.03) before March 2, 2020 (before interventions). A significant linear increment in new confirmed cases was observed, which was about 0 to 1,872.20 (95% CI, 1,257.603 to 2,486.79; p<0.001), after the Nowruz holiday. Furthermore, with a three-day delay, a linear increase in COVID-19 case numbers was observed for eight days after the beginning of the Nowruz holiday.

The results showed no significant change in the number of new confirmed cases after the implementation of the first intervention (CSU). Whereas, after implementing the second intervention (SDM), new daily confirmed cases decreased significantly (p<0.001) from 2,182.80 (95% CI, 1,556.56 to 2,809.04) to 1,343 during the intervention period.

We also modified some parts of the discussion section in lines 220-223, as follows:

The Nowruz holiday (an Iranian national holiday) created a significant increase in the number of COVID-19 cases, three days after its start. Due to the lack of personal protective equipment such as masks and disinfectants, as well as a high level of contact between people, the Nowruz holiday was an occasion for easy transmission of the disease in Iran [28].

We also modified our limitations as follow in lines 262 to 269:

There are some limitations to the analysis conducted in this study. First, we only analyzed the available data related to the period from February 19 to May 2, 2020, as the Iranian Ministry of Health began its active case finding program during this period. Second, we did not have access to sub-groups and geographical-level data. These data could be valuable in determining the heterogeneity of the effect of social distancing in different subgroups and for different geographical areas. Moreover, there were also some variables, such as the level of access to the healthcare system, the changing diagnostic criteria for COVID-19, the people’s compliance to health rules, and preventive programs that could influence the effectiveness of social distancing.

• It would be better if the authors would consider a counterfactual because there is no way to confirm that this epidemic trend will behave the same without these interventions. Most of the time, some areas do not fully accomplish all interventions, and these areas are perfect as a counterfactual.

ANSWER: We agree with the editor’s comment. Although this is interesting, these interventions have been used simultaneously and uniformly everywhere in Iran, so that schools and universities were all closed simultaneously. Nowruz holiday is a national holiday throughout the country, as well as social distancing measures was as they were enforced by the police, and were applied in all the cities of Iran, so it is practically impossible to study the pandemic’s process without considering these interventions. Therefore, a counterfactual cannot be used as a comparison in this case.

• The authors must give information about the pre-emptive measures' compliance to value them and evaluate their effect on the pandemic curve.

ANSWER: Thank you for your comment. Regarding the use of masks as an important pre-emptive measure, during the first months of the pandemic, not only was the use of masks not mandatory, but the number of available masks did not even cover the needs of medical staff. Masks produced by the Ministry of Health were thus collected, rationed, and distributed to medical centers throughout Iran (1).

As the government rejected plans to quarantine entire cities and only urged people to stay at home, this rule was not followed in the early months of the pandemic, and offices and companies remained open, leaving only children, students, and a small number of people at home (2). On April 4th, Iranian officials expressed concerns that many had ignored the rules to stay indoors and to cancel travel plans (3).

Regarding hand washing, it was initially recommended to wear gloves at the beginning of the pandemic, but was later announced that gloves can also lead to transmission, and that it is better to wash hands with soap and water for 20 seconds instead of wearing gloves (4,5). However, most of the population of Iran did not take pre-emptive measures such as wearing a mask, washing hands, and staying at home seriously. We have added a summary of this in the manuscript at lines 95-96 and 221-223.

1. Raoofi A, Takian A, Sari AA, Olyaeemanesh A, Haghighi H, Aarabi M. COVID-19 pandemic and comparative health policy learning in Iran. Arch Iran Med. 2020;23(4):220–34.

2. "Coronavirus: Iran has no plans to quarantine cities, Rouhani says". BBC. 26 February 2020. Archived from the original on 26 February 2020. Retrieved 26 February 2020.

3. "Iranians defy coronavirus rules as death toll reaches 3,452". Arab News. 4 April 2020.

4. https://www.tehrantimes.com/news/446498/COVID-19-crisis-washing-hands-for-20-seconds-not-wasting-water

5. https://www.who.int/gpsc/5may/Hand_Hygiene_Why_How_and_When_Brochure.pdf

1) Reviewer #1: The authors have conducting fairly straightforward analyses using highly aggregated surveillance data with known limitations to answer complex questions. Greater attention needs to be placed in the analytical approach and its appropriateness to the research question. Conceptually, authors need to better address the nature of epidemic growth and reduction, particularly taking into account the limitations of the data used. Finally, the authors make numerous statements attributing causality to their findings without carefully consideration of the causality principles and how their evidence supports them.

ANSWER: We appreciate the reviewer's comments. We have tried to enhance the manuscript by carefully addressing these points. As mentioned above, we have modified the results, discussion, and mentioned data limitations (please see lines 254-259). As for the goal of the interrupted time series, i.e., analyzing the effectiveness of interventions in a dynamic regression framework, the current article’s research question is: To find the effect of three interventions on the daily number of COVID-19 cases. Thus, we have not sought to examine the causal relationship between interventions and the daily number of COVID-19 cases. Therefore, some statements about causality have been corrected in the manuscript in lines 168-182, 191-192 and 237-238.

2) The ARIMA model intrinsically does not have growth restriction and in theory it could grow without limits, which does not correspond to reality, because during epidemics, even in large populations the number of susceptible individuals is eventually exhausted. This is conceptually erroneous when applied to the complete growth and reduction cycle of a pandemic, and it assumes that all the reduction cycle is due to the physical distancing measures, without taking into account the reduction of the susceptible individuals and its impact. It should be clarified if the autocorrelation is the only mechanism that accounts for the multiplicative growth pattern of the epidemic. Particularly because the use of pulse-like effects to estimate the potential effect of the control measures and the Nowruz holidays does not affect the autocorrelation or the slope of the curve.

ANSWER: Thank you for your comment. First, the data used in this study covers the initial 74 days from the beginning of the pandemic. The reduction of susceptible cases is not significant for this period, based on the population of Iran. Also, it should be mentioned that according to the statistics, the number of COVID-19 cases that appeared after the study period were higher, which reinforces our previous argument. Based on this explanation, it is valuable to say that SARIMA can be a good model to measure the pandemic’s growth pattern. It should be noted that the error component in SARIMA modeling explains that time points have an autocorrelation besides pulse-like effects. This is good when there are unknown factors involving the response variable that cannot evaluated, based on limited data and information. 

3) Likewise, it should be clarified if the ARIMA model uses a Poisson or another distribution for count data such as cases, and what are the effects of that in the estimations.

ANSWER: Thank you for your comment. In the ARIMA model, white noise (i.e. errors) suggests a normal distribution which is conventionally assumed in times series. Having normally distributed errors is equivalent to having normally distributed observations for any linear time series model. Although it is not necessary to assume normality of errors, maximum likelihood is often used to estimate the model’s parameters, followed by a Gaussian likelihood, which gives good results even with non-normal data. Normality of errors is often assumed when using the AIC for order selection, and when computing prediction intervals.

4) What is the precision of estimates, and is model fitness high enough that can serve to respond trend questions? No goodness of fit statistics are presented nor are the observed and estimated results presented, visually or in a tabulation.

ANSWER: Thank you for your constructive suggestion. The precision of estimates and the model fitness have been added to the manuscript, as suggested. In this case, the mean absolute percentage error (MAPE) is 1.88, which is a highly accurate forecasting based on the research study by Lewis (1982, p.40). We added to at the end of the results section, in lines 201 to 203.

5) Finally, no sensitivity or sub-group analyses are presented to document the robustness of the results and related conclusions. Most countries have shown highly heterogeneous sub-national epidemic expansions in the first wave of the pandemic, with the national pattern primarily reflecting trends in the capital or largest cities, and very different pattern in smaller or distant/less connected cities. Conclusions could greatly change when addressing these heterogeneities.

ANSWER: Thank you for your comment. There are some limitations to the analysis conducted in this study. First, we only analyzed the data related to the period between February 19 and May 2, 2020, as this is when the Iranian Ministry of Health began its active case detection program. Second, we did not have access to sub-groups and geographical-level data. These data could have been valuable to determine the heterogeneity of the effect of social distancing with different sub-groups and in different geographical areas. Moreover, there were also other variables, such as the level of access to healthcare, changing diagnostic criteria, the people’s compliance to health rules, and preventive programs that could influence the effectiveness of social distancing.

We do understand the reviewer’s concern regarding the possible heterogeneity of the results. However, and as suggested in the limitations section of the manuscript, such a precise analysis cannot be conducted due to the nature of the data, and its availability in Iran.

6) Conceptually, which of the three types of interventions better fits the interventions evaluated, and why were the others considered? Did the empirical results of the regression matched the expectation due to the conceptual type of intervention?

ANSWER: Thank you for your comment. The three types of interventions were evaluated to find the best fit on the data and, based on the nature of each law that is enforced, the type of intervention will be different. So, there is no single answer to the question of what the best type of intervention is. However, in this study, we evaluated the three types of interventions, as explained in the manuscript, to determine which one best fit the data. For instance, the delayed response (linear), and decayed response (exponential) were the best fits for the second and third interventions in the current study. For the second question, it was expected that tCOVID-19 would not have an impact immediately, without delay. Results therefore show that conceived interventions are the same as those hypothesized in the study, based on the nature of COVID-19. 

7) Could the closure of schools have a relatively smaller effect than expected and was not detectable with the methods used? What if this measure slowed down the decrease observed in mid-March 2020?

ANSWER: 

It seems that closing schools has no impact on the number of new COVID-19 cases. This is because the trend of time series before March 20th was not decreasing, it was even showing a growing trend. A review study shows that the school closures during coronavirus outbreaks had a very weak impact on the spread of COVID-19 (1). In other words, the implementation of this law did not reduce the incidence of new coronavirus cases. However, if we assume that without this intervention, the trend may increase more than the current increased rate, it is therefore valuable to say that this method has limitations. So, it is complicated to determine how the trend of time series may change without this intervention and it depends on several variables, especially at the start of the pandemic in Iran, because at this time, it was very difficult to assess all the data and variables surrounding the new COVID-19 cases. 

1-Viner RM, Russell SJ, Croker H, et al. School closure and management practices during coronavirus outbreaks including COVID-19: a rapid systematic review. Lancet Child Adolesc Health. 2020;4(5):397-404. doi:10.1016/S2352-4642(20)30095-X

8) The abstract should present actual results to support the authors’ statements and conclusions, as p-values are insufficient for that purpose.

ANSWER: Thank you for your constructive suggestion. The results section of the abstract was modified in lines 27-32 as follows:

The results showed that, after the implementation of the first intervention, i.e., the closing of universities and schools, no statistically significant change was found in the number of confirmed new cases. The Nowruz holiday was followed by a significant increase in new cases (1,872.20; 95% CI, 1,257.60 to 2,476.79; p<0.001)), while the implementation of social distancing measures was followed by a significant decrease in such cases (2,182.80; 95% CI, 1,556.56 to 2,809.04; p<0.001).

9) The case definition needs to be clearly described stating how testing availability and distribution countrywide affects the validity and proper interpretation of findings

ANSWER: 

In Iran, the first cases of infection were reported in Qom and later spread to other parts of Iran. It should be noted that the following programs were implemented simultaneously with the spread of the pandemic in Iran:

- Launching more than 20 COVID-19 diagnostic laboratories across the country on February 26, 2020. Also, an Iranian Health Ministry spokesman says the Islamic Republic is preparing for the possibility of “tens of thousands” of people showing up to get tested for the new coronavirus on February 29, 2020.

- Increasing the number of coronavirus diagnostic laboratories to 50 centers in Iran on March 5, 2020.

- Launching a self-reporting system to identify suspicious cases and control infected cases on March 11, 2020.

- Launching a new coronavirus detection lab at the Pasteur Institute of Iran and increasing its coronavirus diagnostic capacity to up to 1,800 tests per day on March 14, 2020.(1)

By increasing the number of tests, COVID-19 diagnoses have been increased. It is valuable to mention that the rate of test increment was non-stop, even after social distancing. So, the decrease in COVID-19 patients after this enforced distancing cannot be due to lack of access to testing, or improper distribution. We have added this clarification to the manuscript, in lines 245-249. 

1. Raoofi A, Takian A, Sari AA, Olyaeemanesh A, Haghighi H, Aarabi M. COVID-19 pandemic and comparative health policy learning in Iran. Arch Iran Med. 2020;23(4):220–34. 

10) How many days are expected to pass for interventions to start being effective? How was this lag introduced in the analyses?

ANSWER: Regarding this issue, and for the first question, authors believe that there is no explicit answer concerning the number of expected days to pass for interventions to start being effective. Based on this study, the best lag was found according to the results of statistical modeling. It should be noted that in some cases, there won’t be any significant lags. However, in this paper, the first intervention related to school closings had no effect on new COVID-19 cases, while the Nowruz holiday did have an effect, albeit with a three-day delay. This is because COVID-19 symptoms can take up to 14 days to develop. In addition, for the second question, we analyzed the time series point by point, exactly after each intervention to see whether the intervention had a significant effect on the time series or not.

---

## [Decision Letter · Decision Letter 1]

16 Jul 2021

PONE-D-20-30409R1

Effects of government policies and the Nowruz holidays on confirmed COVID-19 cases in Iran: An intervention time series analysis

PLOS ONE

Dear Dr. Fakoor,

Thank you for submitting your manuscript to PLOS ONE. After careful consideration, we feel that it has merit but does not fully meet PLOS ONE’s publication criteria as it currently stands. Therefore, we invite you to submit a revised version of the manuscript that addresses the points raised during the review process.

Specifically, please address the minor comments of Reviewer 3.

We look forward to receiving your revised manuscript.

Kind regards,

Siew Ann Cheong, Ph.D.

Academic Editor

PLOS ONE

Journal Requirements:

Reviewers' comments:

Reviewer's Responses to Questions

**Comments to the Author**

1. If the authors have adequately addressed your comments raised in a previous round of review and you feel that this manuscript is now acceptable for publication, you may indicate that here to bypass the “Comments to the Author” section, enter your conflict of interest statement in the “Confidential to Editor” section, and submit your "Accept" recommendation.

Reviewer #2: All comments have been addressed

Reviewer #3: (No Response)

2. Is the manuscript technically sound, and do the data support the conclusions?

Reviewer #2: Yes

Reviewer #3: Yes

3. Has the statistical analysis been performed appropriately and rigorously? 

Reviewer #2: Yes

Reviewer #3: Yes

4. Have the authors made all data underlying the findings in their manuscript fully available?

Reviewer #2: Yes

Reviewer #3: Yes

5. Is the manuscript presented in an intelligible fashion and written in standard English?

Reviewer #2: Yes

Reviewer #3: Yes

6. Review Comments to the Author

Reviewer #2: Ref. PONE-D-20-30409R1

Title: Effects of government policies and the Nowruz holidays on confirmed 2 COVID-19 cases in Iran: An intervention time series analysis

The research paper on “Effects of government policies and the Nowruz holidays on confirmed COVID-19 cases in Iran: An intervention time series analysis” is very interesting and well written and used various statistical techniques, I find very useful and appropriate to the general readerships of the journal. The authors have carefully revised the manuscript which was raised by reviewers/ editor. Therefore, I strongly recommend (accepted) this article for publication in the present form.

Reviewer #3: 1. Before proceeding to the rigorous analyses of time series data, the authors need to conduct the basic statistical analysis of the data mentioning the minimum and maximum values, mean, skewness, kurtosis, etc. A diagrammatic representation of the features of the data be made (If possible).

2. The authors should provide autocorrelation function (ACF) and partial autocorrelation function (PACF) plots for the time series under study.

3. ARIMA (Or SARIMA) modeling methodology be briefly explained in the methods.

4. In the Table1, the test-statistic (e.g. t-test or Chi-square test) used need to clearly stated to find the 95% confidence intervals.

5. Could there be any future predictions using the proposed model? It would be best if the proposed model had been capable of drawing future predictions.

7. PLOS authors have the option to publish the peer review history of their article (what does this mean?). If published, this will include your full peer review and any attached files.

Reviewer #2: No

Reviewer #3: No

---

## [Author Response · Author response to Decision Letter 1]

7 Aug 2021

Professor. Siew Ann Cheong

Academic Editor

PLOS ONE 

SUBJECT: Resubmission of manuscript PONE-D-20-30409R1

Re: Effects of government policies and the Nowruz holiday on confirmed COVID-19 cases in Iran: An intervention time series analysis

Dear Editor,

We would like to thank you for the attention brought to our manuscript for this second review. Also, we would like to thank the academic editor and the reviewers for their careful reading of our manuscript and their insightful comments. We have reviewed the comments and, then, reviewed the manuscript accordingly. The comments have been noted and we have done our best to revise the paper. Below is a point-by-point response to each comment raised by the academic editor and the reviewers, especially the third reviewer (original comments from reviewers appear in red, responses in black). It should be noted that the modifications in the manuscript are highlighted using track changes in Word. 

We hope that the revised version of our manuscript will now meet the PLOS ONE’s publication criteria.

Yours sincerely,

Vahid Fakoor

Associate Professor, Department of Statistics, Faculty of Mathematical Sciences, Ferdowsi University of Mashhad, Mashhad-Iran,

P.O.Box, 9177943369 

Mobile: 0989151038414

E-mail: 

Journal Requirements:

ANSWER: Thanks for your comment. We have checked and modified some references (for example 2, 3, 4, 5, 6, 11, 15, 30). Also as you mentioned a paper, that is retracted, is replace with another relevant reference as bellow (in lines 60-63):

11. Zareie B, Roshani A, Mansournia MA, Rasouli MA, Moradi G. A Model for COVID-19 Prediction in Iran Based on China Parameters. Arch Iran Med. 2020 Apr 1;23(4):244-248. doi: 10.34172/aim.2020.05.

1) Reviewer #3: 1. Before proceeding to the rigorous analyses of time series data, the authors need to conduct the basic statistical analysis of the data mentioning the minimum and maximum values, mean, skewness, kurtosis, etc. A diagrammatic representation of the features of the data be made (If possible).

ANSWER: We appreciate the reviewer's comments. We amended them in table 1 in the section method in page 5.

2) The authors should provide autocorrelation function (ACF) and partial autocorrelation function (PACF) plots for the time series under study.

ANSWER: Thank you for your comment. We amended autocorrelation function (ACF) and partial autocorrelation function (PACF) for the residuals of model in page 13, as required. 

3) ARIMA (Or SARIMA) modeling methodology be briefly explained in the methods.

ANSWER: As you suggest, a paragraph explaining these method have been added on page 6 for more clarification on SARIMA modeling as bellow: 

1. SARIMA, firstly, proposed by Box and Jenkins in the 1970s. It is presented as SARIMA (p, d, q)(P, D, Q)S, where p is the order of auto-regressive (AR), q is the order of moving average (MA), d is the order of the differences. The ACF and PACF are used for knowing the order of AR and MA. Also, based on the trend and season of time series, the order of differences will be recognized. Moreover, P, D, and Q are the corresponding seasonal orders, with S as the steps of the seasonal differences (E.P. George, G.M. Jenkins Time series analysis forecasting and control Holden-Day (1976)).

4) In the Table1, the test-statistic (e.g. t-test or Chi-square test) used need to clearly stated to find the 95% confidence intervals.

ANSWER: It should be mentioned that the statistic for obtaining 95% confidence intervals is based on Z test on page 10 and a sentence was added in this regard. A note as also been added under Table 2 (the table number is now 2 with the addition of a new table in the manuscript). 

5) Could there be any future predictions using the proposed model? It would be best if the proposed model had been capable of drawing future predictions.

ANSWER: The capability of SARIMA modeling is in both forecasting and interventional analysis. So, the bellow sentences and table 3 are put in page 13. 

“Table 3 shows the prediction of Covid-19 new confirmed cases, with 95 % confidence interval, in 7 days. It should be noted that the prediction can be utilized if the observed spreading pattern, and the number and type of test for detection the COVID-19 cases continues as before and if policies and restrictions are not removed. Therefore, the number of daily new cases would be 813 and 814 for 3 May 2020 and 4 May 2020, respectively.”

---

## [Editor Report · Decision Letter 2]

10 Aug 2021

Effects of government policies and the Nowruz holidays on confirmed COVID-19 cases in Iran: An intervention time series analysis

PONE-D-20-30409R2

Dear Dr. Fakoor,

We’re pleased to inform you that your manuscript has been judged scientifically suitable for publication and will be formally accepted for publication once it meets all outstanding technical requirements.

Kind regards,

Siew Ann Cheong, Ph.D.

Academic Editor

PLOS ONE
---

## [Editor Report · Acceptance letter]

12 Aug 2021

PONE-D-20-30409R2 

Effects of government policies and the Nowruz holidays on confirmed COVID-19 cases in Iran: An intervention time series analysis 

Dear Dr. Fakoor:

I'm pleased to inform you that your manuscript has been deemed suitable for publication in PLOS ONE. Congratulations! Your manuscript is now with our production department. 

Kind regards, 

on behalf of

Dr. Siew Ann Cheong 

Academic Editor

PLOS ONE